# A bottom-up approach dramatically increases the predictability of body mass from personality traits

**Kadri Arumäe** [1]*, **Uku Vainik** [1,2,3], **René Mõttus** [1,4]

**1** Institute of Psychology, University of Tartu, Tartu, Estonia, **2** Institute of Genomics, University of Tartu, Tartu, Estonia, **3** Montreal Neurological Institute, McGill University, Montreal, Canada, **4** Department of Psychology, University of Edinburgh, Edinburgh, United Kingdom

* kadri.arumae@ut.ee

**Data Availability Statement:** The data were obtained from and are available at https://dataverse.harvard.edu/dataset.xhtml?persistentId=doi:10.7910/DVN/SD7SVE, https://dataverse.harvard.edu/dataset.xhtml?persistentId=doi:10.

## Abstract

Personality traits consistently relate to and allow predicting body mass index (BMI), but these associations may not be adequately captured with existing inventories' domains or facets. Here, we aimed to test the limits of how accurately BMI can be predicted from and described with personality traits. We used three large datasets (combined $N \approx 100,000$) with nearly 700 personality assessment items to (a) empirically identify clusters of personality traits linked to BMI and (b) identify relatively small sets of items that predict BMI as accurately as possible. Factor analysis revealed 14 trait clusters showing well-established personality trait–BMI associations (disorganization, anger) and lesser-known or novel ones (altruism, obedience). Most of items' predictive accuracy (up to $r = .24$ here but plausibly much higher) was captured by relatively few items. Brief scales that predict BMI have potential clinical applications—for instance, screening for risk of excessive weight gain or related complications.

## Introduction

Numerous studies have shown robust relations between personality traits and various health outcomes [1–3], including body weight [4–6]. Body weight matters for life quality in several ways: Not only does excessive weight predict adverse health outcomes [7] and lower subjective well-being [8] but it is also associated with stigmatization and discrimination in many life domains, from education and employment to interpersonal relationships [9]. On one hand, body weight can be predicted from sets of personality traits [10], which could have applications in health care. On the other hand, its descriptive correlations with individual personality traits and trait groups can inform hypotheses regarding the causes and consequences of body weight. To be useful, these predictions and descriptions need to be as accurate as possible. But previous work has not been sufficiently detailed for testing the limits of how accurately BMI can be predicted from and described with personality traits.

Body weight (or body mass index, BMI) has a small but robust negative correlation with Conscientiousness, whereas its relations with the other Five-Factor Model domains have been inconsistent across studies [4,5]. To refine these relations, some studies have considered the

7910/DVN/GU70EV, and https://dataverse.harvard.edu/dataset.xhtml?persistentId=doi:10.7910/DVN/TZJGAT.

**Funding:** UV and KA were funded by Estonian Research Council's (https://etag.ee/en/) grant PSG759. The funder had no role in study design, data collection and analysis, decision to publish, or preparation of the manuscript.

**Competing interests:** The authors have declared that no competing interests exist.

domains' facets, showing that BMI relates not only to several facets of Conscientiousness but also to numerous facets within other domains which themselves may not correlate with BMI [6]. Further necessitating such fine-grained analyses, facets of the same domain sometimes correlate with BMI in different directions. For instance, among Extraversion's facets, BMI correlates positively with Positive Emotions and negatively with Activity [6].

But even items within facets may substantially differ in their correlations with an outcome. For instance, BMI correlates with the Impulsiveness facet of the NEO Personality Inventory (NEO-PI) solely due to its two items reflecting eating behaviour [11,12]; rather than being more impulsive in general, people with overweight or obesity may simply be more prone to overeating. As another example, BMI has no consistent association with Openness to Feelings but correlates with its item reflecting emotional sensitivity to environments [13]. Similar differences in item–BMI associations are likely within other facets [14]. Failing to consider such within-facet heterogeneity can lead to hypotheses about the behavioural contributors to weight gain that are either too unspecific to be useful or entirely incorrect. Likewise, predictive accuracy can be attenuated. We propose that descriptive and predictive analyses could start with individual items as the narrowest units capturing personality differences, only then aggregating these if and as necessary. This bottom-up strategy stands in contrast to the usual top-down approach that starts with trait aggregates and may not even consider their more specific constituents.

We used three large datasets with nearly 700 diverse personality items [15,16] to test the personality trait–body weight associations in a bottom-up way. First, for descriptive purposes, we selected the items that were consistently linked to BMI and aggregated them factor-analytically to bespoke factors (a similar procedure has been employed by Weiss et al [17] for mortality). Although items might be left completely unaggregated for a maximally detailed account of the associations, aggregation enables more parsimonious descriptions. We expected BMI's correlations to be stronger with these empirically derived factors than with the Five-Factor Model domains or facets.

Second, in prediction-focused analyses, we aimed to find optimal item sets for predicting BMI—that is, item sets that are as small as possible but as highly predictive as possible. The current study relies on cross-sectional data and by predicting BMI we mean applying predictive models trained in one sample to other, independent samples. However, there is some evidence that personality items have similar relations to concurrent and future BMI [10], suggesting that the same traits that predict BMI concurrently also predict its future values (perhaps as a result of traits' effects on weight accumulating over time). As personality traits could have useful clinical applications—for instance, they could be used to stratify people based on their risk for adverse outcomes—development of brief but predictively accurate assessment methods may be particularly useful [18], especially for such a common condition like overweight with a range of adverse sequelae. Beyond body weight, BMI also tracks health status more broadly [19], indicating that models that predict BMI also allow predicting health in a more general sense.

To test the robustness of the results, the empirically constructed factors' replicability and the prediction models' accuracy were tested in datasets independent of the ones used to create them.

## Method

### Participants

We used data from three publicly available datasets (total N > 100,000): one collected between the years 2014 and 2015, the second between 2015 and 2017, and the third between 2013 and

2014 [20–22], to which we refer as datasets A, B, and C, respectively. Besides various demographic and health-related variables including BMI, all three datasets contain self-reported information on 696 personality items selected to "include all of the items from a broad and balanced group of measures that were theoretically diverse and frequently referenced in the personality literature" [23]. Various personality inventories measuring the domains of the Five-Factor Model and HEXACO as well as other personality constructs are represented by these items. Many items have some degree of overlap and some are near-identical, ensuring that factors reflecting common content can emerge in factor analysis. The data were collected using the Synthetic Aperture Personality Assessment (SAPA) method which entails administering random subsets of personality items to respondents. Data collected using this method are (massively) missing completely at random [24]; accordingly, the three datasets have 87–88% of missingness in personality variables.

Dataset A included data from approximately 55,000 people, dataset B, 48,000, and dataset C, 23,000. After removing participants with missing information on BMI, age, sex, and country (which we recoded as continent), as well as participants with unlikely values of BMI (BMI $< 14$ or $\geq 75$), 43,237, 37,156, and 19,443 participants remained in datasets A, B, and C, respectively. The samples had a mean age of 26–27 years (range 14–89 in datasets A and B and 14–88 in dataset C) and a mean BMI of 25 kg/m$^2$ with 62–63% of the respondents being female, 39–42% in university or college at the time of responding, and 73–76% located in North America. Sample characteristics of each dataset are further detailed in S1 Table.

## Measures

**Personality variables.** The 696 items included in the SAPA datasets were responded to on a scale from 1 (*very inaccurate*) to 6 (*very accurate*). The Five-Factor Model domains and their facets were assessed using the 300 items that make up the IPIP-NEO which measures each of the five domains with six facets, with each facet consisting of 10 items. We excluded two eating-related items ("Often eat too much" and "Love to eat") belonging to the Neuroticism domain and Immoderation facet of the IPIP-NEO as (over)eating is directly relevant to body weight but not theoretically interesting in the context of the current study. The final item pool thus consisted of 694 items; the Neuroticism domain and Immoderation facet were also calculated excluding these items.

**Body mass index.** BMI (kg/m$^2$) was calculated based on self-reported height and weight.

## Statistical analyses

**Descriptive analysis.** For the descriptive part of the study, we first selected items most relevant to BMI to be included in the analysis. We residualized each of the 694 items and BMI for age, age$^2$, sex, and continent in datasets A and B, and found Pearson's correlations between the residualized items and BMI, selecting items that correlated with BMI at $p < .05$ and in the same direction in both datasets. A lenient $\alpha$ threshold of inclusion was chosen as we preferred a larger number of factors to emerge in the interest of a possibly comprehensive description.

We then constructed correlation matrices with the items selected in the previous step in datasets A and B and conducted exploratory factor analysis (EFA) on these matrices, choosing the number of factors with parallel analysis. Maximum likelihood factoring and oblimin rotation were used. We assigned items to factors based on their primary loadings; items that had no loadings of at least |.30| were discarded. Factors were excluded if they (a) contained fewer than three items, (b) were only found in one of the two datasets, or (c) had no clear interpretation.

Before proceeding with analyses, we assessed the replicability of the factors based on three criteria outlined by Osborne & Fitzpatrick [25] and Condon [23]. First, we ensured that all items loaded on the same factor in both solutions by discarding the items that loaded on different factors. Second, we assessed the equivalence of factor loadings across the datasets by subtracting each item's loading in dataset B from its loading in dataset A and squaring the differences. Osborne & Fitzpatrick [25] suggested that a squared mean difference of $> = .04$ may indicate a volatile factor loading and the respective items deserve further attention. And third, the similarity of the factor loadings in the two solutions was assessed using Pearson's correlation.

**Comparing correlations with different variable types.** To be useful, the empirically constructed factors need to have stronger correlations with BMI than pre-existing traits. Thus, to test their utility, we next compared BMI's correlations with these factors to its correlations with other personality variables: IPIP-NEO domains, facets, and items, as well as various other sets of items: all items in the SAPA datasets, items included in EFA, items included in the factors, items selected for EFA not included in the factors, and the items within the empirically constructed factors with the strongest correlations with BMI. These comparisons would provide context to assess how strongly the empirically constructed factors related to BMI, showing how much of the BMI-relevant information the factors could capture in comparison to other types of personality variables. All items and BMI were residualized for age, age$^2$, sex, and continent prior to constructing the factors, domains, and facets and before calculating the correlations. The factors, domains, and facets were calculated for people who had at least three items in the given factor, domain, or facet available. Datasets A, B, and C were combined to calculate these correlations.

To get a sense of the variable sets' average correlations with BMI, we calculated the mean, median, and range of the absolute values of their Pearson's correlations with BMI. Additionally, we used each set of variables as predictors in a separate elastic net model with BMI as the outcome to assess how accurately the variables within each set were able to collectively predict BMI in an independent sample. Elastic net is a type of penalized regression where the predictors (here: personality variables) are assigned weights based on their intercorrelations to optimize the predictor–outcome associations and thus maximize prediction of the outcome (here: BMI) while minimizing the risk of overfitting [26]. The models were built with 10-fold cross-validation with the α hyperparameter set to .50 and tested with λ chosen to minimize cross-validated error. To further counteract overfitting, we trained and tested the models in independent samples: Datasets A and C were combined for training, and dataset B was used for testing as the training sample ought to be larger than the testing sample for maximum accuracy [27]. Because elastic net models tend to have higher accuracy in larger training samples, we ensured equal training sample sizes for all sets of predictors by limiting the sample to participants for whom at least one domain, factor, and item in each set of predictors was available. We quantified the models' accuracy as Pearson's correlation between the prediction of the elastic net model and actual BMI in the testing sample.

A flowchart of the analyses is shown in Fig 1.

**Prediction optimization.** In prediction-focused analyses, we built elastic net models to maximize predictive accuracy while limiting the number of predictors. We first residualized personality items and BMI for age, age$^2$, sex, and continent. To winnow the predictor pool, iteratively among the 694 personality items combining datasets A, B, and C, the item with the lower correlation with BMI was excluded from the item pair with the highest inter-correlation until none of the remaining items had intercorrelations stronger than a specified criterion. We tried criteria from $r = .01$ to .20 to find sets of predictors that were small enough to be practically useful, but whose predictive accuracy would not have meaningfully increased by the

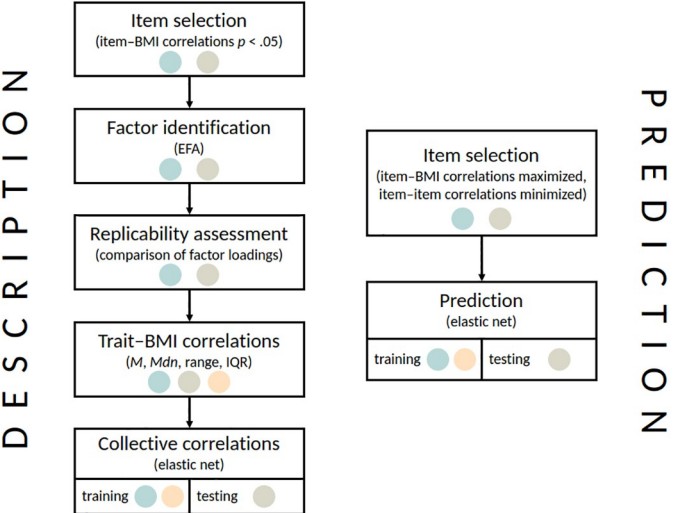

**Fig 1. Analysis flowchart.** The colored dots denote the datasets used in the different steps of the analysis (blue: dataset A, green: dataset B, yellow: dataset C).

addition of another item. Finally, we applied elastic net to test the predictive accuracy of the sets of items (as in descriptive analysis, the models were trained in datasets A and C with 10-fold cross-validation, α set to .50 and tested in dataset B with λ chosen to minimize cross-validation error). We evaluated the models' ability to capture risk for excessive weight by stratifying people in the testing dataset into deciles based on their predicted BMI, expecting people in higher deciles to have higher observed BMI.

All analyses were done with R version 3.6.3 using RStudio.

**Transparency and openness.** The study or the analyses were not preregistered. The data were obtained from https://dataverse.harvard.edu/. Because the present study uses publicly available data, ethical approval was not sought. Following the ethical principles of the American Psychological Association, informed consent was not required as the data used in this study were collected with anonymous online questionnaires that were not expected to distress or harm participants (https://www.apa.org/ethics/code). Analysis code is available at Open Science Framework (https://osf.io/rcdgb/). The scoring key for IPIP-NEO was obtained from https://osf.io/ycvdk/.

## Results

### Descriptive analysis: The factors

Of the 694 items, 242 correlated with BMI at $p < .05$ and in the same direction in both datasets A and B and were thus included in EFA. Parallel analysis indicated 21 factors comprising a total of 120 items in dataset A and 19 factors comprising 131 items in dataset B. We excluded factors only found in one of the two datasets (resulting in a loss of 57 items) and 12 items that loaded on different factors in the two datasets. One factor's items correlated with BMI in opposite directions and were thus divided into two factors called *talkativeness* (items with positive correlations) and *liveliness* (items with negative correlations). One factor (which we labeled Dominance) had a negligible correlation with BMI ($r = .01$) and was excluded from further analyses. Fourteen factors including a total of 82 items remained.

The loadings of three items had squared mean differences exceeding .04 (up to .11) between the two datasets. Because all three items fit well with the other items in the respective factors, we retained all of them. The 82 items' loadings were correlated at $r$ = .83 in the two factor solutions, indicating that the relative importance of the items in the factors was broadly similar across the two datasets.

Table 1 summarizes the final 14 factors and their correlations with BMI in the combined dataset. BMI had the highest correlations with factors like Immoderation, Activity, Disorganization, Anger, Conventionality, Liveliness, and Talkativeness ($|r| \geq$ .08). S2 Table shows the factor loadings of the final 83 items in datasets A and B.

As oblique rotation was used, many of the factors were non-trivially correlated with others (the average absolute correlation was $r$ = .17). Some factors had particularly high correlations with others and with the IPIP-NEO domains (e.g., exceeding $r$ = .50); for example, high correlations were found between the factors of Mood Swings, Worry, and Anger, each of which additionally correlated highly with the Neuroticism domain, and between Liveliness, Talkativeness, and Activity. Correlations between the 14 factors and the domains of IPIP-NEO are shown in S1 Fig.

## BMI's correlations with factors and other personality variables

BMI's correlations with the different types of personality variables are summarized in Fig 2. The 14 factors' and their lead items' average absolute correlations with BMI ($Mdn_{|r|}$ = .08 and $Mdn_{|r|}$ = .07, respectively) were appreciably higher than the rest of the variable sets'. Items of the IPIP-NEO as well as the whole SAPA dataset correlated with BMI at $Mdn_{|r|}$ = .03. The lowest were the correlations of the IPIP-NEO domains ($Mdn_{|r|}$ = .01) and facets ($Mdn_{|r|}$ = .02). Thus, as expected, the empirically constructed factors had stronger correlations with BMI than either the domains or facets.

**Table 1. Summary of the 14 factors.**

| Factor | Number of items | Lead item | N | r (factor) | r (lead item) |
|---|---|---|---|---|---|
| Altruism | 9 | Think of others first. | 15,597 | .05 | .06 |
| Anger | 8 | Lose my temper. | 7,056 | .10 | .11 |
| Obedience | 8 | Try to follow the rules. | 13,777 | .07 | .08 |
| Talkativeness | 8 | Have an intense boisterous laugh. | 7,883 | .08 | .15 |
| Immoderation | 6 | Am able to control my cravings. | 5,572 | -.18 | -.21 |
| Worry | 6 | Panic easily. | 4,386 | .07 | .08 |
| Activity | 5 | Feel healthy and vibrant most of the time. | 4,107 | -.15 | -.16 |
| Conventionality | 5 | See myself as an average person. | 2,762 | .10 | .11 |
| Disorganization | 5 | Leave a mess in my room. | 2,536 | .14 | .16 |
| Liveliness | 5 | Am rather lively. | 3,156 | -.09 | -.11 |
| Mood swings | 5 | Get overwhelmed by emotions. | 2,284 | *.03* | *.04* |
| Adventure-seeking | 4 | Seek adventure. | 2,032 | *-.05* | *-.06* |
| Impulsivity | 4 | Dont know why I do some of the things I do. | 1,251 | *.04* | *.07* |
| Preference for the familiar | 4 | Prefer to stick with things that I know. | 2,428 | .07 | .07 |

The lead item is the factor's item with the highest correlation with BMI. $N$ is the number of people for whom the factor could be calculated (factors were calculated for people for whom at least three items in the factor were available). Correlations were calculated in the combined dataset. Lead items' correlations with BMI were calculated in the same subsample as their respective factors. A correlation coefficient is italicized if statistical power was below .80 for its calculation at the available sample size.

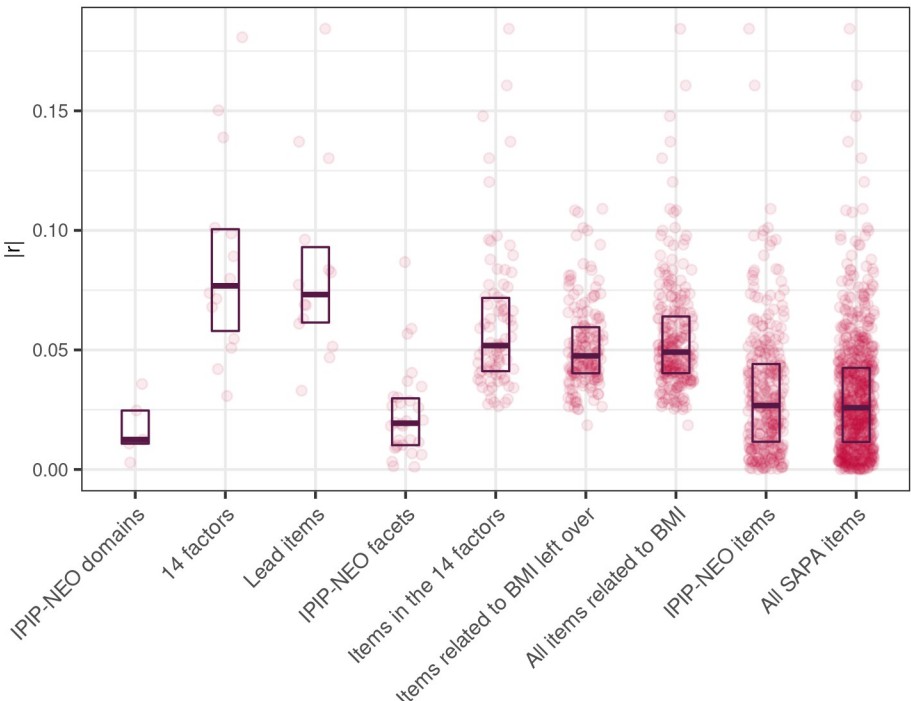

**Fig 2. Variable Set's correlations with BMI.** Dots denote the absolute values of the items' correlations with BMI; crossbars show their medians and interquartile ranges. Variable sets are ordered based on the included number of predictors in the set. Lead items are the factor's items with the strongest correlation with BMI; all items related to BMI include the 242 items with statistically significant links to BMI selected for EFA; items related to BMI left over include items related to BMI that were not included in the factors. The correlations are summarized numerically in S3 Table.

Results of elastic net models are displayed in Fig 3. Predictive accuracy tended to improve as the number of variables within the set of predictors increased: BMI was predicted least accurately by the five domains of IPIP-NEO ($r = .04$) and most accurately by all 694 items as well as the 244 items related to BMI ($r = .28$). This was not because of over-fitting since we trained and tested models in separate samples. Certain exceptions also stood out. The 30 facets of IPIP-NEO had lower accuracy ($r = .07$) than the 15 empirically constructed factors ($r = .12$), and the 300 IPIP-NEO items had lower accuracy ($r = .25$) than the 244 items related to BMI ($r = .28$). Empirically choosing predictors therefore more than doubled the predictive accuracy compared to predefined traits or inventories.

The 244 items related to BMI—those initially chosen for EFA—captured the available BMI-relevant information as adding the remaining 452 items made no improvements. However, the 161 items left over from factors had nearly as high accuracy at $r = .26$: Apparently, the different sets of items captured overlapping predictive signal. The 15 factors' and their lead items' accuracy was considerably lower ($r = .12$ for both).

**Optimizing prediction.** The different criteria applied to item selection resulted in item sets of two to 30 items (S4 Table). Accuracy for the two-item set was $r = .08$. Beyond the two initial items, the largest gain was obtained with a fourth predictor ($r = .11$). The 17-item set achieved an accuracy of $r = .15$; from there on, prediction plateaued and additional items had negligible incremental value. The four- and 17-item sets, which offer a compromise between prediction and number of items, are listed in Table 2 along with their weights in elastic net

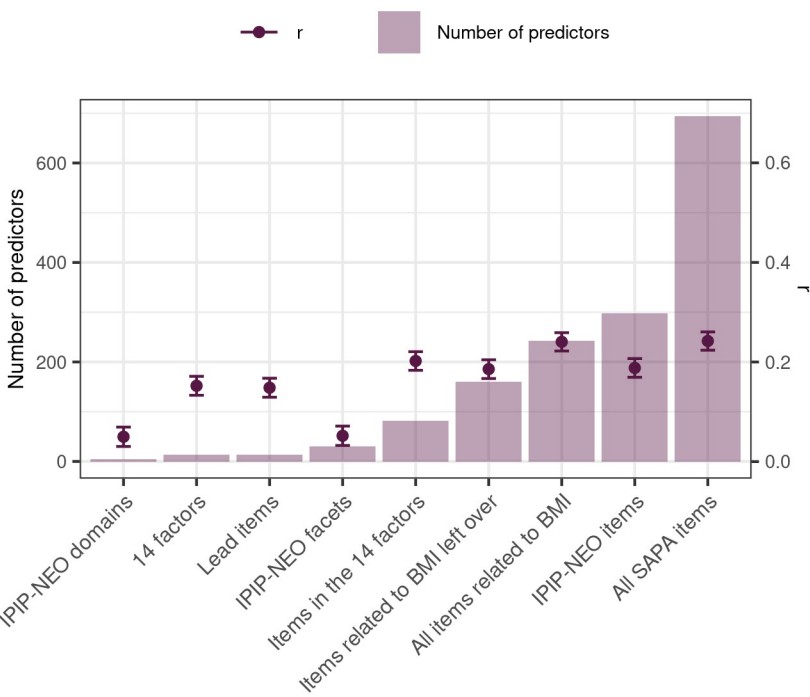

**Fig 3. Variable sets' correlations With BMI in elastic net.** *r* marks the correlation between actual BMI and BMI as predicted by the elastic net model with the given set of personality variables; error bars indicate 95% confidence intervals. The results are shown numerically in S3 Table.

models; the items comprising the remaining sets are listed in S5 Table. Altogether, the results suggest that most of the prediction can be attained with relatively few items: Compared to the maximum prediction (*r* = .24, obtained using the 242 items with the strongest links to BMI; see the previous section), the four items captured 46% of the signal using just 2% of the items and the 17 items captured 63% of the signal using 7% of the items.

The four- and 17-item sets' ability to capture risk for excessive weight is illustrated in Fig 4. Both had some ability to differentiate people with the highest and lowest risk: People in the highest decile were 1.40 kg/m$^2$ or 5.08 kg heavier than people in the lowest decile as stratified by the four-item model and 2.91 kg/m$^2$ or 8.40 kg heavier as stratified by the 17-item model. Although the middle predicted deciles did not differentiate well between body weight levels, both models thus had some ability to identify people with the highest risk for obesity.

## Discussion

BMI is among the most readily assessable and available health outcomes and its high levels have a wide range of adverse downstream consequences for both health and well-being. Accurately describing its psychological correlates can help to better understand its causes and consequences, while accurately predicting it can help to identify those at risk. We used a large and diverse pool of personality items to explore personality traits' associations with BMI in more detail than the domains and facets of standard personality inventories have allowed so far. Aggregating the most relevant items factor-analytically, we found higher BMI related to higher scores on immoderation, disorganization, anger, and conventionality, and lower scores on activity and liveliness, to name a few. Despite the numerous correlations, relatively small sets of items captured most of the predictive signal found in the nearly 700 items. Given these

**Table 2. The four- and 17-item predictive scales.**

| Item | Elastic net weight |
| --- | --- |
| Four items ($r_{BMI}$ = .11) | |
| Am able to control my cravings. | -0.831 |
| Have an intense, boisterous laugh. | 0.409 |
| Wish to stay young forever. | -0.053 |
| Never go down rapids in a canoe. | 0.254 |
| 17 items ($r_{BMI}$ = .15) | |
| Am able to control my cravings. | -0.767 |
| Am just an ordinary person. | 0.389 |
| Believe that we coddle criminals too much. | 0.339 |
| Can't do without the company of others. | -0.067 |
| Cry during movies. | 0.103 |
| Don't strive for elegance in my appearance. | 0.406 |
| Enjoy feeling "close to the earth." | -0.228 |
| Have an intense, boisterous laugh. | 0.417 |
| Like music. | 0.112 |
| Like to take it easy. | 0.450 |
| Love my enemies. | 0.160 |
| Seldom toot my own horn. | -0.088 |
| Wish to stay young forever. | -0.064 |
| Never go down rapids in a canoe. | 0.225 |
| Yell at people. | 0.314 |
| Often take notice of what people think. | -0.031 |
| Worry about my health. | 0.324 |

results, it is possible to create bespoke scales of relatively few items that maximize the prediction of BMI and could have potential applications in precision medicine.

## BMI's links with empirically constructed traits

Aggregating the items with the strongest relations to BMI, 14 factors emerged and their factor structures and correlations with BMI were replicated across the datasets. These factors were generally narrower in content than the domains of standard inventories like the NEO-PI [28], but many conceptually aligned with their facets: For instance, the factors Altruism, Worry, Adventure-seeking, Talkativeness, and Impulsivity all have plausible analogues in the NEO-PI. These empirically constructed factors could be expected to relate to body weight as their analogous facets have been previously reported to correlate with BMI. Indeed, we found this to be the case for some factors: The NEO-PI analogues of the factors we labeled Anger, Impulsivity, and Activity had similar correlations with BMI in a previous meta-analysis [6].

For many similarly labeled traits, however, the associations diverged from those reported previously. For instance, we found BMI to correlate with Altruism, Worry, Talkativeness, and Adventure-seeking, but none of these traits' NEO-PI facet analogues have consistent meta-analytic relations with BMI [6]. The likely explanation is that the items in the predefined facets have different relations to body weight. Further, some factors had no analogues among the NEO-PI facets (e.g., Mood Swings or Obedience) and vice versa: Some traits that correlated with BMI in earlier facet- and item-based analyses [6,14] had no analogous factors in the current results (e.g., Positive Emotions, Assertiveness, or Self-consciousness), likely because these scales' BMI-correlations were driven by only some items.

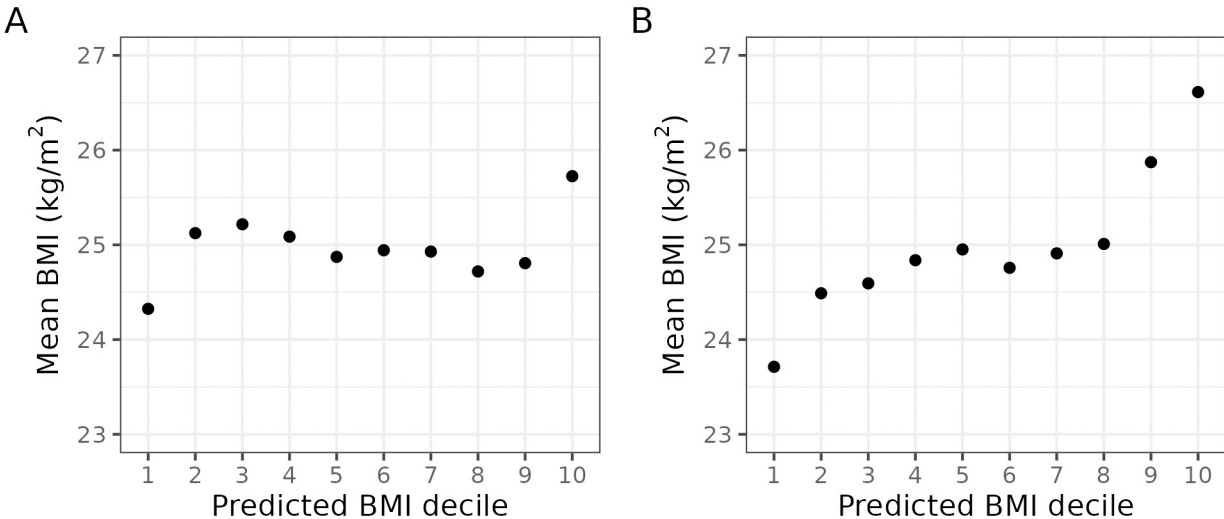

**Fig 4. Relationship between predicted and observed BMI.** Predictions from the 4-item (panel A) and 17-item models (panel B). The results are shown numerically in S6 Table.

Regardless of the different analytic approach, the current results agree with previous ones in many ways. Namely, people with higher body weight have been reported to be more impulsive and prone to anger, less organized and active, and have a higher preference for routine [4,6,14], which the current results also suggest. Although BMI has generally not been related to worry (the facet of neuroticism) [6], obesity has nevertheless been associated with symptoms of anxiety disorders [29]. The current results suggest that such tendencies can also be detected with personality items.

We also detected several associations that are not commonly reported. People with higher BMI were, on average, more altruistic, adventure-seeking, conventional, obedient (or rule-abiding), and talkative while also being less lively. Some of these relations seem to be at odds with one another. For instance, people with higher BMI were more altruistic but also angry; they were more talkative but did not tend to talk to many people at parties or enjoy being a part of a group (items within Liveliness). However, these links may be only seemingly contradictory as each correlation may be driven by a different subset of people—the people with high BMI who are altruistic are not necessarily the same ones who are prone to anger. As others have noted [30], there may not be a single personality profile associated with over- or underweight; higher and lower weight may instead accompany many different configurations of traits.

As for effect sizes, different types of traits did correlate with BMI with different strengths. The IPIP-NEO domains' correlations were weak ($|r|$ = .00–.04) and their facets' correlations somewhat stronger (.00–.09). Compared to the analogous NEO-PI traits, the IPIP-NEO domains' associations were in a similar range and their facets' associations slightly higher [6]. Importantly and as expected, though, the empirically constructed factors' correlations were stronger (.03–.18) than either the domains' or facets' correlations on average, indicating that personality traits in general are more relevant to body weight than the domains and facets have shown. This account of trait–BMI associations can also provide input for new causal hypotheses as well as aid in finding them as stronger correlations can be detected more easily.

Combining sets of personality variables to predict BMI, we similarly found that the IPIP-NEO domains and facets were outpredicted by alternative predictor sets. That facets'

accuracy was no higher than the domains' ($r$ = .05 for both) was somewhat surprising—there were six times as many facets as there were domains and they should be better able to capture the lower-level associations. The relatively low accuracy of facet models may be due to the facets' items differing substantially in their associations with BMI. The empirically constructed factors and their lead items performed better than both facets and domains ($r$ = .15), suggesting item-specificity in BMI-associations, and a larger number of items generally meant more accurate prediction (we emphasize again that our findings did not result from statistical overfitting because we cross-validated them across independent subsamples). Yet, not all items improved accuracy: The whole item pool's prediction accuracy, $r$ = .24, was also achieved by the set of 242 items with statistically significant links to BMI.

To provide some context for this result, several studies have used personality items in similar machine learning models to predict BMI with accuracies generally not exceeding $r$ = .25 [10,31,32], but being $r$ = .43 at maximum when using a pool of 135 items [27]. Relevant to the current results, the latter study also tested prediction at different imposed levels of data missingness, finding prediction with complete personality data to be 269% higher than at 90% missingness. In view of that, prediction could also be considerably higher with the current study's item pool with low or no missing data, potentially exceeding the accuracy obtained in previous studies.

## Optimizing prediction

Our search for brief item scales that predict BMI as accurately as possible yielded four- and 17-item sets with predictive accuracies of $r$ = .11 and $r$ = .15, respectively. Although the correlations between predicted and measured BMI may seem modest by traditional standards, the respective weight differences between people with the highest and lowest predicted risk were over 5 and 8 kg for the four- and 17-item scales. Importantly, as discussed in the previous section, the same models could feasibly obtain accuracies of over 2.5 times higher given complete data [27]; the 17-item set could thus come close to the accuracy of the strongest prediction reported to date (i.e., $r$ = .43 at 135 items) [27].

Because the current results were obtained with cross-sectional data, conclusions regarding the prediction of weight gain should be made cautiously. Nevertheless, if similar prospective associations are shown to exist, brief self-report scales based on the item sets reported here could have potential clinical applications [32]. For instance, stratifying people based on risk of excessive weight gain could help efforts to prevent obesity and its downstream consequences [18]. Whereas the four- and 17-item models did not clearly differentiate people in the middle deciles of predicted risk, they performed better in classifying people with the highest and lowest risk, suggesting some power to identify people with elevated (and reduced) risk for excessive weight.

Besides personality traits, various biological and environmental variables can predict body weight. One study found parental socioeconomic status to be more predictive of BMI than 27 narrow personality traits in adolescents [33], suggesting that the predictive utility of such systemic variables as parental education, income, and occupational prestige may be higher than that of many individual-level variables. Another study found a genetic score calculated from over 2 million genetic variants to predict BMI with an accuracy of $r$ = .29, translating to a 13 kg gradient between the highest and lowest predicted BMI deciles [34]. Although the genetic score's accuracy is nearly twice as high as the prediction provided by the brief personality scales, this does not undermine the utility of personality-based prediction. On the contrary: Genetic data are expensive and burdensome to obtain but responding to 17 personality items

—whose predictive accuracy might realistically be considerably higher—takes two minutes and no money.

Whereas each type of predictor has its unique advantages and limitations in comparison to others in terms of the cost and effort of obtaining and predictive signal, it is likely that the most accurate predictions can be obtained when combining several sources of data. For instance, doubt has been expressed as to whether genetic prediction models will ever be able to accurately predict obesity on their own, given the condition's aetiological complexity [35]. Moreover, different types of predictors could capture partially overlapping predictive signal as, for example, genetic variants could affect body weight partly through personality traits and socioeconomic factors. Still, each type of variable likely adds unique information; for instance, socioeconomic status and narrow personality traits each have some unique signal in predicting BMI in adolescents [33]. Thus, what type of data best predicts weight gain is not the only important question; the party interested in making the predictions could also consider what data are available or can realistically be obtained to reach a predictive accuracy they consider sufficient.

## Limitations

First and foremost, the current study was limited by the availability of large datasets with broad representations of personality traits. Although the data of the SAPA project included a broad set of items, its items have been selected to cover many commonly measured constructs but may exclude others. In addition, some constructs may have been represented with only single items and therefore not aggregated into factors. Hence, body weight may correlate with additional traits not reported here.

Second, BMI was calculated from self-reported height and weight, which tend to be less accurate than their objective measurements and can introduce bias. For instance, people with certain personality characteristics could underreport body weight, which could suppress the traits' correlations with BMI [36]. Third, BMI can conflate lean mass and fat mass which can bias its associations with personality and other variables [37]; yet, BMI tracks fat mass closely on the population level [38], suggesting its suitability for population-level predictive modeling. Fourth, high data missingness limited statistical power in calculating the factors' correlations with BMI and likely affected predictive accuracy as predictions are considerably more accurate with complete data [27]. Fifth, the data were cross-sectional and did not enable conclusions about possible longitudinal associations—but given that longitudinal associations between personality traits and BMI are roughly similar to cross-sectional ones [10], it is likely the same traits could predict future BMI with similar accuracy as they can concurrent BMI. And finally, although the sample was geographically diverse, the analyses were not focused on cultural differences, but different traits may be linked to body weight between cultures [39] and the results may not generalize to all populations.

## Conclusions

Consistent with the expectation that personality traits' associations with body weight are more specific than can be described with standard personality inventories, we found bottom-up, item-based approaches to be more useful than domain- or facet-based ones for both descriptive and predictive purposes. In descriptive analyses, body weight was linked to 14 clusters of traits. Some of the results replicated well-known associations (like with anger and disorganization) while other links have not been found using traditional approaches (like with altruism and obedience). In predictive analyses and under the restrictive condition of 90% data missingness, sets of personality items predicted BMI with accuracies up to $r = .24$, but most of this

predictive signal could be captured with relatively few items. Brief assessment scales based on these item sets could be useful in precision medicine—for instance, in identifying people likely to develop high BMI and thus be at risk for its various downstream consequences.

## Supporting information

**S1 Table. Sample characteristics.**
(ODT)

**S2 Table. Items included in the factors.**
(ODT)

**S3 Table. Summary of personality variables' correlations with BMI.**
(ODT)

**S4 Table. Accuracy of predicting BMI at different criteria of interitem correlations.**
(ODT)

**S5 Table. The predictive scales.**
(ODT)

**S6 Table. Observed BMI and Weight by Predicted BMI Decile.**
(ODT)

**S1 Fig. Correlations between the 15 factors and IPIP-NEO domains.** The factors and domains were calculated after residualizing the items for age, $age^2$, sex, and continent. Factors and domains were calculated for people for whom at least three items in the factor (domain) were available.
(PNG)

## Author Contributions

**Conceptualization:** Kadri Arumäe.

**Formal analysis:** Kadri Arumäe.

**Funding acquisition:** Uku Vainik.

**Investigation:** Kadri Arumäe.

**Methodology:** Kadri Arumäe, René Mõttus.

**Software:** Kadri Arumäe.

**Supervision:** Uku Vainik, René Mõttus.

**Visualization:** Kadri Arumäe.

**Writing – original draft:** Kadri Arumäe.

**Writing – review & editing:** Kadri Arumäe, Uku Vainik, René Mõttus.

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
