## [Decision Letter · Decision Letter 0]

22 Aug 2023

PONE-D-23-06586A bottom-up approach dramatically increases the predictability of body mass from personality traitsPLOS ONE

Dear Dr. Arumäe,

Thank you for submitting your manuscript to PLOS ONE. After careful consideration, we feel that it has merit but does not fully meet PLOS ONE’s publication criteria as it currently stands. Therefore, we invite you to submit a revised version of the manuscript that addresses the points raised during the review process.

We look forward to receiving your revised manuscript.

Kind regards,

Sohaib Mustafa

Academic Editor

PLOS ONE

Journal Requirements:

Reviewers' comments:

Reviewer's Responses to Questions

**Comments to the Author**

1. Is the manuscript technically sound, and do the data support the conclusions?

Reviewer #1: Yes

Reviewer #2: Partly

2. Has the statistical analysis been performed appropriately and rigorously? 

Reviewer #1: Yes

Reviewer #2: No

3. Have the authors made all data underlying the findings in their manuscript fully available?

Reviewer #1: Yes

Reviewer #2: Yes

4. Is the manuscript presented in an intelligible fashion and written in standard English?

Reviewer #1: Yes

Reviewer #2: Yes

5. Review Comments to the Author

Reviewer #1: I found this manuscript ("A bottom-up approach dramatically increases the predictability of body mass from personality traits") to be clear, concise, and insightful. The introduction was breezy and easy to read; the methods were robust and well-considered. The results were generally straight-forward, if a bit complicated (see comments below). The discussion is thorough and, in my opinion, quite thoughtful.

I have only a few suggestions for the authors to consider. The first relates to the analyses for Figure 3. The text about this was somewhat confusing, and it caused me to wonder whether the main takeaway needed so much elaboration. Can it be streamlined? I think it's useful for readers to know that the best-subsets prediction plateaus (at .14-.16 with 15 items), and that relatively few items do most of the work. If retained, the figure itself would benefit from better labeling.

In the following paragraph, I found myself wanting to see an evaluation of the extent to which predicted values (e.g., decile means) relate to observed values. Unless I'm misunderstanding, this was not given. The importance of the results that are presented only became clear to me when I got to the interpretation in the Discussion (page 18). In that section, the comparison to genetic prediction was interesting and worth keeping.

The last 2 suggestions will make me seem a bit vain, but I feel the need to mention two publications of my own that are relevant for the current work -- I've given the full references below. The first is straight-forward to explain. The authors currently cite Condon & Revelle (2015), as is appropriate for the data from 2013-2014. This is dataset C in the current analyses. But, the data from 2014-2015 (Dataset A) and 2015-2017 (Dataset B) are not described in that same paper. Instead, they are described in Condon, Roney, & Revelle (2017), so I think this should be added. If the authors want to streamline the references a bit, I think it would be fine to drop the references currently labeled #18-20. I don't think these add much info beyond what is discussed in the others except that they provide direct links to the data (so, maybe they should be kept).

Finally, there is a recently published paper that is complementary to the findings reported here -- Weston, Leszko, & Condon (2023). It also focuses on BMI and personality using a more recent sample from the same data source (SAPA; the data used for that work are not yet open), but it looks only at adolescents in the United States. The research questions and findings are not overlapping, but it seems likely that researchers seeking to follow up on the work reported here may benefit from a cross-reference. Along those lines, I think the main takeaway of Weston et al. (2023) really should be raised in the Discussion section (perhaps around line 375). Specifically, the finding is that personality is not nearly as predictive of overweight and obesity in adolescents as socioeconomic status (SES). In my view, the case for personalized medicine and intervention based on personality factors is made more credible when contextualized relative to other constructs. This point is beyond the scope of the current work, but I think it would be useful to acknowledge the idea someplace in it.

- line 126, Whe  We

- line 233, "inclearing" should maybe be "the included"

- reference #10 (Arumäe, Mõttus, & Vainik) is missing some details

Condon, D. M., Roney, E., & Revelle, W. (2017). A SAPA project update: On the structure of phrased self-report personality items. Journal of Open Psychology Data, 5:3, https://doi.org/10.5334/jopd.32

Weston, S., Leszko, M., & Condon, D. (2023). Body mass in US adolescents: Stronger ties to socioeconomic status than personality. Personality Science, 4, 1-24. https://doi.org/10.5964/ps.7703

Signed,

David Condon

Reviewer #2: This paper aims to empirically (bottom-up) construct factors predictive of BMI across three datasets. There is a lot to like about this study: the large samples and personality item coverage; the thoroughness of the description of the analyses; the transparency and focus on replicability. However, there some conceptual and methodological concerns that dampen my enthusiasm for the manuscript in its present form.

One issue I have is of contamination of BMI-relevant content in the personality items. In other words, some - maybe many - of the personality items have content that directly or indirectly speaks to or flows from their BMI (e.g., I worry about my health) - the obvious reason these items are coming up has nothing to do with personality and everything to do with weight. In other words, too much of the association with BMI and the item pool, I think, is carried by the 'contamination' of the personality domains by content that directly concerns weight, physical health, or physical activity, so in the end, personality isn't really being used to predict the criterion (or not as much as they would like to think). In that case, I might argue it might just be simpler (for a clinician) to ask for height and weight directly. I would suggest combing through the items that might directly or indirectly speak to BMI and do a sensitivity test – whether the items that you derive once you remove the BMI-relevant content are still useful.

The R-squared the authors report is quite low suggesting in turn low clinical utility in personalized medicine or otherwise. This needs to be explicitly stated in the limitations – the idea that using personality items to predict BMI is far from being prime-time ready.

The study is cross-sectional, but the authors’ language has some implications about prospective functioning. Since the authors don’t have any way of determining directionality references to it should be removed.

I wasn’t sure which analyses used which datasets (separately or together) - a figure or flowchart might help

It would be helpful to discuss the limits of BMI as a measure of obesity/speaking to health somewhere in the discussion.

6. PLOS authors have the option to publish the peer review history of their article (what does this mean?). If published, this will include your full peer review and any attached files.

Reviewer #1: **Yes: **David M Condon

Reviewer #2: No

---

## [Author Response · Author response to Decision Letter 0]

6 Oct 2023

We thank the Reviewers for their thoughtful comments. After considering their suggestions, we clarified some aspects of our methods and results and refocused our discussion on the potential clinical utility of the predictive models. We also restructured the discussion to better differentiate the descriptive and predictive analyses’ results. Additionally, we made some minor changes throughout the manuscript for readability.

We respond to the Reviewers’ comments below.

Reviewer #1: I found this manuscript ("A bottom-up approach dramatically increases the predictability of body mass from personality traits") to be clear, concise, and insightful. The introduction was breezy and easy to read; the methods were robust and well-considered. The results were generally straight-forward, if a bit complicated (see comments below). The discussion is thorough and, in my opinion, quite thoughtful.

I have only a few suggestions for the authors to consider. The first relates to the analyses for Figure 3. The text about this was somewhat confusing, and it caused me to wonder whether the main takeaway needed so much elaboration. Can it be streamlined? I think it's useful for readers to know that the best-subsets prediction plateaus (at .14-.16 with 15 items), and that relatively few items do most of the work. If retained, the figure itself would benefit from better labeling.

In the following paragraph, I found myself wanting to see an evaluation of the extent to which predicted values (e.g., decile means) relate to observed values. Unless I'm misunderstanding, this was not given. The importance of the results that are presented only became clear to me when I got to the interpretation in the Discussion (page 18). In that section, the comparison to genetic prediction was interesting and worth keeping.

Answer: We rewrote the Optimizing prediction section with clarity of the message in mind (lines 277–292) and added a new figure to illustate how predicted BMI related to observed BMI (Fig 4). To clarify the relevance of stratifying participants into deciles early on we now also mention it in the methods (lines 191–193).

Reviewer #1: The last 2 suggestions will make me seem a bit vain, but I feel the need to mention two publications of my own that are relevant for the current work -- I've given the full references below. The first is straight-forward to explain. The authors currently cite Condon & Revelle (2015), as is appropriate for the data from 2013-2014. This is dataset C in the current analyses. But, the data from 2014-2015 (Dataset A) and 2015-2017 (Dataset B) are not described in that same paper. Instead, they are described in Condon, Roney, & Revelle (2017), so I think this should be added. If the authors want to streamline the references a bit, I think it would be fine to drop the references currently labeled #18-20. I don't think these add much info beyond what is discussed in the others except that they provide direct links to the data (so, maybe they should be kept).

Answer: A reference to Condon, Roney, & Revelle (2017) has been added (line 66). We think the most transparent practice is to include links to the datasets and thus kept the existing references.

Reviewer #1: Finally, there is a recently published paper that is complementary to the findings reported here -- Weston, Leszko, & Condon (2023). It also focuses on BMI and personality using a more recent sample from the same data source (SAPA; the data used for that work are not yet open), but it looks only at adolescents in the United States. The research questions and findings are not overlapping, but it seems likely that researchers seeking to follow up on the work reported here may benefit from a cross-reference. Along those lines, I think the main takeaway of Weston et al. (2023) really should be raised in the Discussion section (perhaps around line 375). Specifically, the finding is that personality is not nearly as predictive of overweight and obesity in adolescents as socioeconomic status (SES). In my view, the case for personalized medicine and intervention based on personality factors is made more credible when contextualized relative to other constructs. This point is beyond the scope of the current work, but I think it would be useful to acknowledge the idea someplace in it.

Answer: We agree that Weston et al’s (2023) results help contextualize our own and have added the reference (lines 394 and 411).

Reviewer #1: 

- line 126, Whe  We

- line 233, "inclearing" should maybe be "the included"

- reference #10 (Arumäe, Mõttus, & Vainik) is missing some details

Condon, D. M., Roney, E., & Revelle, W. (2017). A SAPA project update: On the structure of phrased self-report personality items. Journal of Open Psychology Data, 5:3, https://doi.org/10.5334/jopd.

Weston, S., Leszko, M., & Condon, D. (2023). Body mass in US adolescents: Stronger ties to socioeconomic status than personality. Personality Science, 4, 1-24. https://doi.org/10.5964/ps.

Signed,

David Condon

Answer: The errors have been corrected.

Reviewer #2: This paper aims to empirically (bottom-up) construct factors predictive of BMI across three datasets. There is a lot to like about this study: the large samples and personality item coverage; the thoroughness of the description of the analyses; the transparency and focus on replicability. However, there some conceptual and methodological concerns that dampen my enthusiasm for the manuscript in its present form.

One issue I have is of contamination of BMI-relevant content in the personality items. In other words, some - maybe many - of the personality items have content that directly or indirectly speaks to or flows from their BMI (e.g., I worry about my health) - the obvious reason these items are coming up has nothing to do with personality and everything to do with weight. In other words, too much of the association with BMI and the item pool, I think, is carried by the 'contamination' of the personality domains by content that directly concerns weight, physical health, or physical activity, so in the end, personality isn't really being used to predict the criterion (or not as much as they would like to think). In that case, I might argue it might just be simpler (for a clinician) to ask for height and weight directly. I would suggest combing through the items that might directly or indirectly speak to BMI and do a sensitivity test – whether the items that you derive once you remove the BMI-relevant content are still useful.

Answer: While we agree that “contamination” is an important issue in causal analyses, it is not necessarily an issue in predictive analyses such as the ones presented in our current manuscript. Given our objective to assess the limits of how strongly BMI can be predicted from personality traits, it makes sense to include as many and diverse items reflecting personality as possible. Worrying about one’s health may be a likely consequence of excess weight but it seems unlikely that no variation remains between people in the extent that they worry about their health after accounting for weight (for instance, so-called healthy neuroticism; e.g. Friedman, 2019, http://doi.org/10.1037/per0000274). For this reason, we decided not to omit this or other items that could intuitively be contaminated with body weight.

Reviewer #2: The R-squared the authors report is quite low suggesting in turn low clinical utility in personalized medicine or otherwise. This needs to be explicitly stated in the limitations – the idea that using personality items to predict BMI is far from being prime-time ready.

Answer: With this comment as well as Reviewer #1’s comment about comparative predictive accuracy in mind, we have refocused the discussion of the predictive models’ potential utility (lines 376–414). Briefly: each data source (genetics, environmental factors, individual-level psychosocial factors like personality traits) has its strengths and limitations and, while the predictive signal from any one source likely partially overlaps with others, best prediction is likely achieved when combining multiple sources. Even if traits’ predictive utility is low, they could still provide incremental prediction.

Reviewer #2: The study is cross-sectional, but the authors’ language has some implications about prospective functioning. Since the authors don’t have any way of determining directionality references to it should be removed.

Answer: In addition to the limitations section (lines 430–433), we now mention the limitation of cross-sectional data in the introduction (lines 74–79) and the general discussion (lines 384–385). Although we refrain from making any causal interpretations, the utility of predictive models, in the clinical context or otherwise, is independent of whether or not causal influences exist between the predictors and the outcome.

Reviewer #2: I wasn’t sure which analyses used which datasets (separately or together) - a figure or flowchart might help

Answer: A flowchart of the analytic process has been added (Figure 1) as well as some clarifications in the text (lines 157–158 and 184).

Reviewer #2: It would be helpful to discuss the limits of BMI as a measure of obesity/speaking to health somewhere in the discussion.

Answer: The caveat that BMI cannot differentiate between fat mass and lean mass has been added to the limitations section (lines 425–427), acknowledging that this may affect individual-level and sample-level analyses differently. We think it is also worthwhile to mention that BMI is not just an index of body weight but also health more generally; this has been added to the introduction (lines 83–84).

---

## [Decision Letter · Decision Letter 1]

21 Nov 2023

A bottom-up approach dramatically increases the predictability of body mass from personality traits

PONE-D-23-06586R1

Dear Dr. Mõttus,

We’re pleased to inform you that your manuscript has been judged scientifically suitable for publication and will be formally accepted for publication once it meets all outstanding technical requirements.

Kind regards,

Sohaib Mustafa

Academic Editor

PLOS ONE

Additional Editor Comments (optional):

Reviewers' comments:

Reviewer's Responses to Questions

**Comments to the Author**

1. If the authors have adequately addressed your comments raised in a previous round of review and you feel that this manuscript is now acceptable for publication, you may indicate that here to bypass the “Comments to the Author” section, enter your conflict of interest statement in the “Confidential to Editor” section, and submit your "Accept" recommendation.

Reviewer #2: (No Response)

2. Is the manuscript technically sound, and do the data support the conclusions?

Reviewer #2: Yes

3. Has the statistical analysis been performed appropriately and rigorously? 

Reviewer #2: Yes

4. Have the authors made all data underlying the findings in their manuscript fully available?

Reviewer #2: Yes

5. Is the manuscript presented in an intelligible fashion and written in standard English?

Reviewer #2: Yes

6. Review Comments to the Author

Reviewer #2: The authors addressed most of my concerns. The only one that remains unaddressed is the sensitivity analyses – what would happen if you took all personality items relevant to activity, health concerns, etc out. The authors chose not to run these analyses; I don’t feel strongly enough about this to keep making the point, and I’ll make the editor make that call. Other than that, it looks fine.

7. PLOS authors have the option to publish the peer review history of their article (what does this mean?). If published, this will include your full peer review and any attached files.

Reviewer #2: **Yes: **Marina A Bornovalova

---

## [Editor Report · Acceptance letter]

20 Dec 2023

PONE-D-23-06586R1 

PLOS ONE

Dear Dr. Mõttus, 

I'm pleased to inform you that your manuscript has been deemed suitable for publication in PLOS ONE. Congratulations! Your manuscript is now being handed over to our production team.

Kind regards, 

on behalf of

Dr. Sohaib Mustafa 

Academic Editor

PLOS ONE